# Ceruloplasmin Deamidation in Neurodegeneration: From Loss to Gain of Function

**DOI:** 10.3390/ijms22020663

**Published:** 2021-01-11

**Authors:** Alan Zanardi, Massimo Alessio

**Affiliations:** Proteome Biochemistry, IRCCS-Ospedale San Raffaele, 20132 Milan, Italy; alessio.massimo@hsr.it

**Keywords:** ceruloplasmin, choroid plexus, blood-cerebrospinal fluid barrier, NGR and isoDGR motifs, deamidation, oxidation, neurodegeneration, cerebrospinal fluid

## Abstract

Neurodegenerative disorders can induce modifications of several proteins; one of which is ceruloplasmin (Cp), a ferroxidase enzyme found modified in the cerebrospinal fluid (CSF) of neurodegenerative diseases patients. Cp modifications are caused by the oxidation induced by the pathological environment and are usually associated with activity loss. Together with oxidation, deamidation of Cp was found in the CSF from Alzheimer’s and Parkinson’s disease patients. Protein deamidation is a process characterized by asparagine residues conversion in either aspartate or isoaspartate, depending on protein sequence/structure and cellular environment. Cp deamidation occurs at two Asparagine-Glycine-Arginine (NGR)-motifs which, once deamidated to isoAspartate-Glycine-Arginine (isoDGR), bind integrins, a family of receptors mediating cell adhesion. Therefore, on the one hand, Cp modifications lead to loss of enzymatic activity, while on the other hand, these alterations confer gain of function to Cp. In fact, deamidated Cp binds to integrins and triggers intracellular signaling on choroid plexus epithelial cells, changing cell functioning. Working in concert with the oxidative environment, Cp deamidation could reach different target cells in the brain, altering their physiology and causing detrimental effects, which might contribute to the pathological mechanism.

## 1. Introduction

The occurrence of aberrant post-translational modifications in brain proteins is a feature reported in different neurodegenerative diseases, including Amyotrophic Lateral Sclerosis (ALS), Alzheimer’s (AD) and Parkinson’s (PD) disease [1,2,3,4]. These modifications generally involve proteins directly related to the pathology, such as superoxide dismutase, TDP-43 protein, Aβ peptide, Tau protein and α-synuclein [1,2,3,4] but are not limited to them.

Post-translational modifications of ceruloplasmin (Cp), a ferroxidase enzyme present inter alia in the brain, have been observed in the cerebrospinal fluid (CSF) of patients suffering from neurodegeneration [5,6,7,8]. The Cp modifications are usually associated with loss of enzymatic activity and are promoted by the pathological environment [5,6,7,8]. One of the Cp modifications, found in the CSF from both AD and PD patients, is deamidation [9,10], a spontaneous process connected to protein aging [11]. Protein deamidation occurs on two amino acid residues, namely asparagine and glutamine, and generally results in loss of protein function [11]. In the case of Cp, the deamidation involves asparagine residues at the level of two asparagine-glycine-arginine (Asn-Gly-Arg, NGR) sites of the protein sequence whose, upon deamidation, gain binding activity to integrins [9,10,12]. Integrins are a family of receptors that mediated cell-extracellular matrix adhesion [13]. The binding of modified Cp to integrins triggers in vitro intracellular signaling on epithelial cells, altering the cell physiology with the induction of detrimental effects [9,12]. However, despite the presence of NGR motifs deamidation in the Cp from neurodegenerative diseases patients’ CSF, in vivo cellular targets and possible pathological effects are still unexplored topics.

## 2. Protein Deamidation, Unwanted Phenomenon or Regulated Process?

Asparagine (Asn, N) deamidation is a spontaneous reaction in which the amide functional group of Asn is removed, and the amino acid is converted to either an aspartate (Asp, D) or an isoaspartate (isoAsp, isoD) residue [11]. Glutamine (Gln, Q) can also be subjected to deamidation, but at lower rate compared to Asn [14]. It has been demonstrated that deamidation is a post-translational modification that spontaneously occurs in vivo, under physiological conditions [15]. The process is autocatalytic and nonenzymatic, and it depends on the protein primary, secondary, and tertiary structure [16]. The rate of deamidation is also determined by the cellular environment, for example by temperature, ionic composition and strength, and pH changes [11].

Through deamidation, Asn forms a cyclic succinimide intermediate, which is then hydrolyzed to either Asp or isoAsp; due to the asymmetry of the succinimide, isoAsp is the primary product (Figure 1). The formation of isoAsp can also arise from dehydration of Asp, but at slower rate than Asn residues [17]. Although in a smaller proportion, the racemization of succinimide can even result in D-Asp and D-isoAsp [17,18]. Depending on the combination of the conditions mentioned above (sequence, structure, temperature, etc.), the spontaneous deamidation of specific residues can take different times to happen: for some residues, it will occur very quickly, while for some others it will never occur [11,19].

Asn deamidation is irreversible, but the isoAsp formed by the reaction can be further modified by the protein L-isoaspartyl methyltransferase (PIMT) enzyme, which converts isoAsp into Asp. Therefore, PIMT is considered a sort of repair system that limits the protein structural changes fostered by the presence of isoAsp, and indeed, due to this role, PIMT is present ubiquitously in tissues and is widely expressed in Bacteria, Archaea, and Eukarya [20].

Initially, the deamidation was thought to be purely a sort of protein damage associated to aging [18,21]. However, since both Asn and Gln residues show such instability, but are widely distributed in nature, it was proposed that deamidation can serve as a biological timer for protein turnover [21]. In fact, Asn deamidation rate depends on the amino acid sequence, so Asn deamidation could be genetically programmed.

## 3. Deamidation of Asn Residue in NGR Motifs: From Loss- to Gain-of-Function

Since deamidation introduces a negative charge and, in the case of isoAsp formation, changes the length of the peptide bond, this modification can induce protein loss of function. Due to local charge and conformational alterations, the functional interaction with other proteins or substrate molecules could be prevented [11,22]. For instance, in rat calmodulin the deamidation reduces the activity below 20% of the native protein [23], whereas the deamidation of the histidine-containing protein HPr impairs phosphohydrolysis and phosphotransferase activity in Bacteria [24].

Nevertheless, in some cases, Asn deamidation can result in gain of function. For example, it has been demonstrated that Asn deamidation in fibronectin is associated with increased integrin-binding properties [25]. Integrins are cell-adhesion receptors which recognize different extracellular matrix (ECM) proteins [13]. Integrins can bind multiple ligands through different specific recognition sequences [13]. One of these is the Arginine-Glycine-Aspartate (RGD) motif present in some ECM proteins like fibronectin, vitronectin, bone sialoprotein, collagen, and thrombospondin [13]. However, it has been reported that integrins can recognize another sequence within the same RGD-binding pocket: the isoDGR motif resulting from Asn deamidation of the NGR motif [26]. Indeed, structural studies showed that isoDGR binds αVβ3 integrin in reverse orientation compared to RGD, thus maintaining the tertiary structure suitable for fitting within the binding pocket [27]. It has been observed that the deamidation of Asn-263 at the NGR site of fibronectin leads to αVβ3 integrin-binding activity [25,27]. Therefore, Asn deamidation and the subsequent production of isoAsp can act as a molecular switch for integrin-binding [22].

NGR motifs are also present in other proteins. A search in the Swiss-Prot databases found that about 5.02% of proteins contain the NGR-motif, but the frequency of NGR in proteins classified with the keyword “adhesion” is 17.23%. In comparison, the more investigated RGD motif is present in 6.35% of proteins [28]. We performed an up-date of this research for NGR motifs in UniProtKB/Swiss-Prot database (release 2020_05 of 7 October 2020, 563,552 entries), using Prosite tool in the Expasy bioinformatics resource portal (www.expasy.org). Excluding splice variants and restricting the analysis to the *Homo sapiens* database, we found that 5.9% of proteins contain the NGR motif in their sequence (1193 out of 20,385). Within the total NGR-containing proteins, most of them have one NGR motif (92%), while only 8% of the proteins have two or more NGR motifs (Figure 2).

For comparison, the integrin-binding RGD motif was found to be present in 7.7% of all human proteins, and within these, the presence of two or more RGD motifs is restricted to the 7.6%. NGR sites are therefore relatively infrequent in proteins and having two or more NGR motifs is a rare circumstance (only 0.47% of total human proteins). Performing a Gene Ontology (GO) biological process enrichment analysis by using the STRING database (www.string-db.org), we found that 14.5% of the proteins classified as to be involved in cell adhesion, contain NGR motifs; indeed, up to 18.0% of the proteins localized in the extracellular matrix hold one or more NGR site (GO Cellular Component) [29,30].

## 4. Ceruloplasmin Deamidation in Neurodegenerative Diseases

NGR motifs are not only present in adhesion proteins but also in proteins classified as “receptor” and “enzyme” [28]. Ceruloplasmin (Cp) is one of the enzymes that contain NGR motifs. Cp is a multicopper ferroxidase which is mainly synthesized by hepatocytes and secreted into the blood [31]; within the central nervous system (CNS), Cp is expressed by astrocytes as glycosylphosphatidylinositol (GPI)-anchored isoform [32], and is secreted in CSF by the epithelial cells of choroid plexus facing the brain ventricles [31,32,33,34,35]. In particular, Cp is within the restricted group of proteins containing two NGR motifs: the first (^568^NGR) is exposed on the protein surface, while the second (^962^NGR) is hidden within the protein tertiary structure [9]. Having two or more NGR motifs is rare in proteins, and in the case of Cp their importance is underlined by the fact that the two NGR motifs are conserved across different species [10]. It has been observed that Asn in the two NGR motifs carried by Cp can undergo deamidation.

The first ^568^NGR motif deamidates when exposed to conditions that accelerate Asn aging [9], as for the in vitro incubation at 37 °C in ammonium bicarbonate buffer at basic pH [25]. The deamidation results in a local structural change that flip-out the ^568^NGR loop toward the external milieu [9]. On the contrary, the second ^962^NGR motif, buried in the depth of the protein structure, can deamidate only under particular conditions. Since the Asn-962 is locked in a stable conformation between two β-strands, its deamidation occurs only after Cp structural changes that “open” the region where the ^962^NGR site is located. Such conditions may arise when Cp is exposed to an oxidative environment. Indeed it has been reported that exposing Cp to accelerated aging in vitro under oxidative conditions (e.g., in the presence of hydrogen peroxide) promotes the deamidation of Asn at position 962 [9]. Structural alterations induced by oxidation on Cp have been described resulting in the release of coordinated copper atoms [36] and in changes of circular dichroism spectra [9]. Under these conditions, the structural changes and the release of copper atoms leads to the loss of Cp ferroxidase function [8,10]. In addition, the deamidation of Cp induces the conversion of NGR sites into isoDGR, promoting integrin-binding properties and, therefore, a gain of function. Of note, the acquisition of integrin-binding properties happens also when purified Cp is incubated in the CSF from Alzheimer’s disease (AD-CSF) patients or Parkinson’s disease (PD-CSF) patients, but not when incubated in the CSF from control peripheral neuropathy patient and healthy subjects [9,10]. Thus, in the CSF of AD and PD patients are present compounds that foster Cp structural changes and deamidation, suggesting that Cp alterations can also occur in vivo on the endogenous protein. In fact, the Cp present in PD-CSF has been reported to be more oxidized and deamidated than the Cp in the CSF of healthy subjects [8,10]. Nevertheless, in contrast to what was observed in vitro, Asn deamidation in the Cp from PD-CSF was found predominantly at the level of the hidden ^962^NGR site rather than at the level of the more exposed ^568^NGR [10]. This may be explained by the fact that Asn deamidation is dependent on the combination of various elements, among which ionic strength and composition, and pH [11]; therefore, the kinetic of deamidation of a specific Asn residue can change in considerable amount in different conditions. The increased amount of deamidation observed in Cp resident, or incubated, in PD-CSF could be explained by the pro-oxidant environment generated by the hydrogen peroxide, present at high concentration in PD-CSF, compared to healthy subjects and control groups (50 μM vs. 20–25 μM, respectively) [10]. The hypothesis that hydrogen peroxide concentration in the pathological CSF might be responsible for Cp oxidation and deamidation is further supported by the evidence that incubation with catalase, a scavenger enzyme for hydrogen peroxide, prevented Cp deamidation [10]. Since Cp concentration in the CSF ranges from 0.8 to 2.2 μg/mL [37,38], based on quantitative analysis performed by mass spectrometry it has been estimated that deamidated Cp in patients might reach concentration of about 200 ng/mL [10].

The oxidative changes of Cp have double effects: on one hand, the induction of structural change with the release copper atoms coordinated within the Cp catalytic pocket [39], which results in loss of Cp ferroxidase activity [36]; on the other hand, it allows the NGR motifs deamidation, leading to Cp integrin-binding gain-of-function.

## 5. Ceruloplasmin Deamidation and Switch to Integrins Binding Function

As mentioned before, oxidized and deamidated Cp acquires integrin-binding properties which in turn mediate cell adhesion and activate integrin-mediated intracellular signaling pathways. Cp oxidized and deamidated (Cp-ox/de) in vitro binds different integrins, among which αVβ6, whose expression is restricted to epithelial cells [40]. Indeed, it has been reported that Cp-ox/de mediated HaCat epithelial cell line adhesion and spreading onto a coated plate [9]. Moreover, Cp-ox/de induced a series of protein phosphorylation events related to integrin-mediated signaling transduction, particularly those regulating MAPK signaling pathway and cell cycle [9]. The Cp-ox/de interaction with integrins caused cell proliferation arrest due to cell cycle blocking and apoptosis triggering, affecting the epithelial cell functionality [12]. The same effects of cell-adhesion promotion and proliferation inhibition were also observed with Cp modified ex vivo, by the incubation in the CSF from PD or AD patients, suggesting that cellular targeting by modified Cp might also occur in vivo [9,10,12]. These effects on HaCat epithelial cell functionality were mediated by the direct binding to integrins through the deamidated NGR motifs of Cp, as inferred by competitive binding with a peptide containing the isoDGR motif and by Cp-ox/de pre-treatment with PIMT enzyme, which convert the isoDGR integrin-binding motif to DGR, thus avoiding integrins engagement [9,10,12,20,41].

However, HaCat cells are keratinocytes, a type of epithelial cells not present in the CNS and not in contact with the CSF. In vivo, within the CNS the brain ventricle wall is lined by a layer of epithelial cells, namely the ependymal and the choroid plexus cells, possible targets of deamidated Cp. While ciliated ependymal cells lining the brain parenchyma seem to play a principal role in CSF flow through the brain ventricular system, choroid plexus epithelial cells (CPEpiCs) are responsible for the production and the secretion of about 70% of the CSF [42,43]. Indeed, CPEpiCs contribute to the formation of CSF in its fundamental components by: (1) exchanging water, salts, metal ions, metabolites, and proteins from the bloodstream to the ventricles, with the support of a variety of pumps, channels and transporters; (2) secreting metabolites and proteins directly within the CSF [44,45]. Recently, it has been demonstrated that CPEpiCs also release extracellular vesicles (microvesicles and exosomes) into the CSF, and this has been proposed/hypothesized as a cell communication mechanism between blood and CNS [46,47]. Moreover, the epithelial cells of the choroid plexus are responsible for the generation and maintenance of the blood-cerebrospinal fluid barrier (BCSFB) that, in analogy with the blood-brain barrier, separates the CSF from the blood [43].

The possibility to target CPEpiCs is supported by evidence that, via integrin-binding, Cp-ox/de promotes cell adhesion of primary human choroid plexus epithelial cells (HCPEpiCs), which express αVβ3 and α5β1 integrins suitable for isoDGR binding, transducing an intracellular signaling that, as for HaCat, inhibited cell proliferation [12]. This might have consequences on the major physiological roles of these cells, namely secretion and barrier properties, which contribute to determining the CSF composition. For example, the inhibition of CPEpiCs proliferation might limit the cell self-renewal necessary to maintain an intact cell monolayer required for barrier properties. In turn, a barrier leakage can affect both the turnover of CSF and its composition modifying the concentration of metabolites, salts, metal ions and proteins. In addition, since CPEpiCs have high proteins secretion capacity, as both proteins and microvesicles/exosomes [46,48], an aberrant intracellular signaling might also modify the profile of the released proteins. In line with this hypothesis is our observation that treatment with Cp-ox/de is able to induce alterations in the secretome profile of HCPEpiCs [49]. Within 470 proteins identified and quantified, 25 resulted to be differentially expressed and most of them (23 out of 25) showed a reduction in the expression levels compared to untreated cells, while two proteins were up-regulated. These proteins comprise proteins involved in basal membrane and ECM organization, proteases, protease inhibitors, proteins associated to integrins functionality, to cell proliferation, neuronal function and neurodegeneration [49]. Therefore, integrins engagement by Cp-ox/de seems to affect the cellular interaction with ECM and the homeostasis of brain extracellular milieu. However, it must be considered that integrins are adhesion molecules present at the basolateral membrane side of the epithelial cells, while the modified Cp is present in the CSF which soaks the apical side of the choroid plexus. Alteration of the BCSFB properties, which might allow the engagement of the CPEpiCs integrins by modified Cp, has been reported in AD, PD and ALS. These alterations include modification in choroid plexus thickness, mechanical properties and barrier leakage [50,51,52,53,54]. One responsible for the induction of the impairment of BCSFB properties in the neurodegenerative disease is the pro-oxidant environment of the pathological CSF, which is able to induce tight junctions dysregulation and disassembly in epithelial cells [55]. In choroid plexus, as in any other physiological barrier, tight junctions formation and integrity are fundamental for the generation and the maintenance of the BCSFB [43]. The high hydrogen peroxide concentration found in PD-CSF can exert is detrimental effect both by inducing CSF proteins modifications and by dysregulating tight junctions organization [8,10,55]. The same oxidative stress directly affects the Cp structure causing the release of the coordinated copper atoms [36]. Since Cp, which contains six copper atoms, is the major transporter of this metal ion within serum and CSF [31], the local free copper concentration reached upon release from oxidized Cp might contribute to barrier damage. Indeed, copper induced alteration of the barrier permeability has been observed in Caco-2 epithelial cell line [56]. In addition, copper modulates modifications and aggregations of proteins involved in neurodegeneration (e.g., α-synuclein and amyloid-β), affecting the choroid plexus barrier properties [57,58,59,60,61]. The release of copper atoms from Cp leads to loss of ferroxidase activity, as they are fundamental for the electron-transport necessary for the catalytic activity [62], and a decrease of Cp activity in the CSF from both AD and PD patients has been reported as a consequence of the modification induced by the pro-oxidant environment [5,6]. The lack, or decrease, of Cp-mediated ferroxidase activity promotes the intracellular accumulation of reduced ferrous iron, which may react with hydrogen peroxide generating reactive oxygen species through Fenton chemistry [63], which in turn may contribute to the barrier damage. As exemplified by the rare genetic disease aceruloplasminemia [64,65,66], Cp ferroxidase activity is important for the correct iron metabolism, and its absence contributes to oxidative stress, as observed in the brain of both patients and animal models [67,68]. Besides, loss of Cp ferroxidase activity induces iron accumulation in CPEpiCs [69], which could also worsen barrier damage.

In this scenario, the brain oxidative stress, characterizing different neurodegenerative diseases as AD, PD, and ALS [70,71,72,73], could act in different ways, converging on Cp and the choroid plexus. Indeed, oxidative compounds present in the pathological CSF could increase permeability at the level of the BCSFB; in the meanwhile, oxidative stress modifies Cp, promoting its deamidation and gain of integrin-binding functions while reducing its ferroxidase activity. With increased barrier permeability, the modified Cp can cross the epithelial monolayer of the choroid plexus, reaching integrins on the opposite site, where it could exert its detrimental effects on both cell proliferation and secretome composition (Figure 3).

## 6. Asparagine Deamidation and Isoaspartyl Formation in Brain Proteins

In the light of the proposed model of action, it is evident that Cp could not be the unique protein in CNS that undergoes deamidation in pathological conditions acquiring new detrimental functions. This is underlined by the PIMT knock-out mouse neurological phenotype, characterized by widespread brain protein deamidation and fatal seizure [74]. As mentioned before, PIMT enzyme counteracts isoAsp formation [20], thus could regress isoDGR formation, reverting its integrin-binding activity. However, none of the proteins found to be the principal target of PIMT, in the mouse brain, contain NGR motifs in their sequences [75]. Indeed, for the majority of these proteins, the detrimental role of isoAsp formation seems to be due to loss of protein activity rather than gain of function, as exemplified by the loss of enzymatic activity of creatine kinase B after isoaspartyl formation [76]. It should be noted that Asn deamidation does not seem detrimental per se, rather than isoAsp formation that might cause changes in charge and length of the peptide bond, promoting loss of functions [11,22]. However, Asn deamidation was found to increase the propensity to form aggregates of different proteins related to neurodegenerative diseases, among which α-synuclein [77], superoxide dismutase [78], Tau protein [79], amyloid-β [80], therefore providing gain of function.

Nevertheless, other proteins possessing NGR motif/s within the sequence could deamidate to isoDGR due to brain pro-oxidant environment, acquiring integrin-binding properties as seen for Cp. For example, performing the previously mentioned bioinformatics analysis, we found that in proteins related to synapse organization (a biological process exclusively confined to the nervous system), 15.3% display NGR motifs in their sequence, whereas no one has RGD motifs. So, it is possible that some of these proteins might contribute to pathological alterations with a mechanism similar to that observed for Cp. In particular, considering that integrins recognized by isoDGR motif (such as αVβ3, αVβ5, αVβ6,αVβ8 and α5β1) [9,22,25] are widely expressed in the nervous system by both neuronal and glial cells [81], other cells out of CPEpiCs could also be targeted by the NGR-containing deamidated proteins. In addition, variations in the susceptibility of the cells that could be targeted by deamidated proteins may occur as consequences of integrin pattern expression changes fostered by oxidative stress [82].

Therefore, the role of isoAsp formation at the level of NGR motifs in extracellular proteins not related to cell adhesion, and their gain of integrin-binding activity, could be a new field of study which might be worth exploring; especially considering that the neurodegenerative environment itself seems to promote these protein fate changes.

## 7. The Physiological Role of Cp NGR Motifs and Their Deamidation

A question that is still open is the physiological role of the NGR motifs embedded within protein sequences. NGR-containing peptides, both synthetic and physiologically generated (by proteins fragmentation as observed for fibronectin [22,25]), are able to bind the membrane receptor CD13 [41,83]; however, no evidence of NGR-mediated interaction of intact proteins with CD13, or any other receptor, have been so far reported. Thus, it looks like that the molecular switch from NGR to isoDGR, induced by protein deamidation, can be the physiological fate of this motif. Until now, this interchange seems to occur mostly under pro-oxidant pathological conditions, like those present in the CSF of neurodegenerative diseases but also in atherosclerotic plaques where deamidation of NGR motifs have been reported to occur on the ECM proteins promoting integrin-mediated monocyte adhesion [84]. Remarkably, the deamidation of the ^962^NGR motif in Cp has also been observed in another pro-oxidant environment, namely the serum of type 2 diabetes patients [85]. Whether the consequences of NGR-to-isoDGR switch are prevalently detrimental or favorable is not known at the moment. From the evolutionary point of view, it is hard to believe that the NGR motif was maintained within a restricted number of proteins if it predominantly plays a harmful role. The existence of the “repair” enzyme PIMT might suggest the presence of a regulated event that controls integrin-binding activity and signal transduction, which in turn affects cell adhesion and proliferation. Indeed, at least for Cp, the NGR deamidation is also detectable in the CSF of healthy subjects, although in a smaller fraction than is observed in PD patients [10]. Since deamidation is connected to protein aging, this might represent the physiological event marking aged Cp protein targeted for removal and/or degradation [11]. Alternatively, the rate of isoDGR formation in Cp might affect the balance between its physiological versus pathological role of transducing signal to epithelial cells. In physiological conditions, the integrin-mediated cell proliferation inhibition and apoptosis induction might contribute to epithelial cell renewal; while under pathological conditions, the increased rate of isoDGR production in Cp could force this effect toward more aberrant signaling in which apoptosis prevails. An example of this is anoikis, which is a form of apoptosis induced by inappropriate cell-matrix interaction [86,87].

Finally, from both physiological and pathological points of view, it remains to be defined whether the two NGR motifs carried by Cp sequence play similar, synergistic or alternative roles.

## 8. Conclusions

The observation that such a relatively rare motif, which has the potential to bind integrins once modified by the extracellular environment, is present at the level of a secreted protein, like ceruloplasmin, is very intriguing. Whether this phenomenon is part of a physiological pathway still remains unclear. We suggested that deamidation occurring in brain proteins could not only have the effect of loss of function, but also could induce gain of toxic functions, as observed for ceruloplasmin. Further studies are necessary to clarify the impact of brain protein deamidation in neurodegenerative diseases.

## Figures and Tables

**Figure 1 ijms-22-00663-f001:**
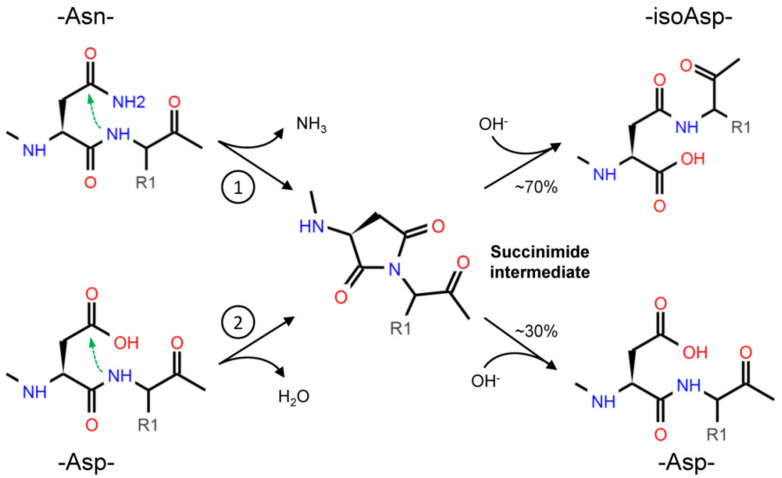
Asparagine (Asn) deamidation (1) starts with the nucleophilic attack (green arrow) of the α-amino group, in the peptide bond, to the amide group in the side chain. The cyclic succinimide intermediate is then rapidly hydrolyzed to a mixture of Aspartate (Asp) and isoAspartate (isoAsp), with the prevalent production of the latest (about 70% vs. 30%) due to the asymmetry of the succinimide [17]. The formation of isoAsp can also occur from Asp dehydration (2), with a similar nucleophilic attack, but this reaction occurs at a slower rate.

**Figure 2 ijms-22-00663-f002:**
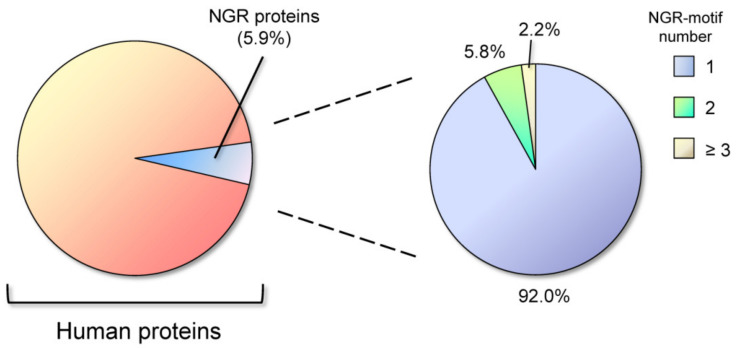
Distribution of Asparagine-Glycine-Arginine (NGR) motifs in human proteins. About 6% of total proteins contain the NGR motif, and the vast majority of these (92%) has one NGR motif in the sequence, while only 8% shows two or more NGR motifs.

**Figure 3 ijms-22-00663-f003:**
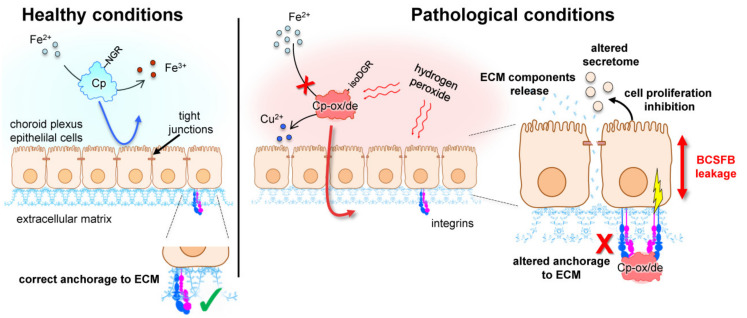
Schematic representation of the possible mechanism of action of oxidized/deamidated ceruloplasmin (Cp-ox/de) on choroid plexus epithelial cells. In healthy conditions, the blood-cerebrospinal fluid barrier (BCSFB) is intact and ceruloplasmin (Cp) is retained in the cerebrospinal fluid (CSF), exerting ferroxidase activity. Under pathological conditions, Cp is modified by the pro-oxidant environment that promotes Asparagine-Glycine-Arginine (NGR) motifs deamidation and loss of copper ions, inhibiting Cp ferroxidase activity. The oxidative environment also induces the leakage of the BCSFB, and allows Cp-ox/de to cross the choroid plexus epithelial monolayer. Once in the basal side, Cp-ox/de binds integrins via isoAspartate-Glycine-Arginine (isoDGR) motif, transducing an intracellular signal that might affect integrin-mediated interaction with the extracellular matrix (ECM), modify secretome components profile, and induce cell proliferation inhibition.

## Data Availability

Not applicable.

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
