# Peer review of "Ceruloplasmin Deamidation in Neurodegeneration: From Loss to Gain of Function"

_ijms, 2021, doi:10.3390/ijms22020663_

Round 1

Reviewer 1 Report

Overall, this manuscript makes a good impression. The authors discuss the consequences of deamidation of two NGR-motifs in ceruloplasmin, which is related to the pathogenesis of neurodegenerative diseases. Unfortunately, when formatting the manuscript, the references lost their numbers, so I'm not sure I identified them correctly. In my opinion, the authors should provide their own data or literature data on the concentration of ceruloplasmin and its deamidated forms in the cerebrospinal fluid.

On page 6 (first paragraph), there is probably a misstake in the dimension of hydrogen peroxide concentration. As far as I remember, 5 mM of H2O2 completely destroy ceruloplasmin due to the site-specific generation of hydroxyl radicals (Sokolov A.V. et al. (2012) Protection of ceruloplasmin by lactoferrin against hydroxyl radicals is pH dependent. Biochem Cell Biol. V. 90(3), P. 397-404. doi.org/10.1139/o2012-004).

I consider it possible to publish this manuscript after eliminating the small remarks indicated above.

Author Response

Response to Reviewer 1 Comments

Point 1: Unfortunately, when formatting the manuscript, the references lost their numbers, so I'm not sure I identified them correctly.  

Author’s response: We apologize for the inconvenience, we have provided for the correction of the mistake.

Point 2: In my opinion, the authors should provide their own data or literature data on the concentration of ceruloplasmin and its deamidated forms in the cerebrospinal fluid.

Author’s response: Data regarding both ceruloplasmin concentration and its deamidated forms in CSF have been added at page 6 – lines 226 to 230: “Since Cp concentration in the CSF ranges from 0.8 to 2.2 μg/ml [37,38], based on quantitative analysis performed by mass spectrometry, it has been estimated that deamidated Cp in the CSF of PD patients might reach concentration of about 200 ng/ml [10].”

Point 3: On page 6 (first paragraph), there is probably a mistake in the dimension of hydrogen peroxide concentration.

Author’s response: As the Reviewer noticed, there was a mistake in the dimension of the concentration due to typos. The correct values are in the range of μMolar instead of mMolar. The correction has been uploaded in the revised manuscript (page 6 – line 221).

Reviewer 2 Report

In this review, the authors describe the link between neurodegenerative disorders and ceruloplasmin deamidation. Authors specifically focused on the dual role of ceruloplasmin modifications, which can lead to loss of enzymatic activity or confer gain of function to ceruloplasmin.

The review is concise, well written and interesting.

In general, I have a few comments and suggestions to the authors:

  1. Abstract – The sentence “…signaling on epithelial and choroid plexus cells,…”, should be corrected to … signaling on choroid plexus epithelial cells…

  1. Ceruloplasmin deamidation in neurodegenerative diseases

- The authors mentioned that the ceruloplasmin is secreted in the CSF by the choroid plexus epithelial cells. More details should be provided about the data corroborating such affirmation.

- The sentence “… Cp is incubated in the CFS from…” should be corrected to … Cp is incubated in the CSF from….

  1. Ceruloplasmin deamidation and switch to integrins binding function

- Is there any evidence of the expression of αVβ6 in the choroid plexus epithelial cells? Is there any idea of possible integrins expressed in the choroid plexus that could bind to deamidated Cp?

- The authors mentioned that the possibility to target choroid plexus epithelial cells might have consequences on the CSF composition. It should be interesting to specify the consequences. CSF turnover? CSF protein concentration?

- The alterations induced in the secretome profile of the choroid plexus epithelial cells after treatment with Cp-ox/de should be discussed in more detail. Treatment with Cp-ox/de increase or decrease the secretion of proteins? Which ones? The treatment induced the release of additional proteins?

  1. References

- The authors should include in the reference list the respective number of each reference mentioned in the manuscript.

Author Response

Response to Reviewer 2 Comments

Point 1:  Abstract – The sentence “…signaling on epithelial and choroid plexus cells,…”, should be corrected to … signaling on choroid plexus epithelial cells… 

 Author’s response: The sentence has been accordingly modified “... signaling on choroid plexus epithelial cells...”.

Point 2: The authors mentioned that the ceruloplasmin is secreted in the CSF by the choroid plexus epithelial cells. More details should be provided about the data corroborating such affirmation

Author’s response: The literature regarding ceruloplasmin and choroid plexus epithelial cells is scanty, in particular the data showing ceruloplasmin protein expression. Most of the expression data are referring to ceruloplasmin mRNA detection performed with primers that are not suitable to discriminate between the membrane GPI-anchored and the secreted isoforms. However, the paper from Patel et al. (J Biol Chem, 272: 20185-90, 1997) shows, by immunofluorescence, the absence of GPI-anchored ceruloplasmin isoform expression on the membrane of choroid plexus epithelial cells, therefore the detected mRNAs are referring to the secreted isoform. This reference, which was already quoted in our originally submitted manuscript (number 32), has been now introduced also in the sentence “..., and is secreted in CSF by the epithelial cells of choroid plexus facing the brain ventricles [31-35].” (page 5, line 172), together with two new references showing ceruloplasmin mRNA expression in the choroid plexus [references 34 (Aldred et al. 1987) and 35 (Rouault et al. 2009)].

In addition, for the Reviewer evaluation only, we attach our unpublished data (please, see cover letter attachment) in which we observed that primary human choroid plexus epithelial cells express ceruloplasmin and secrete the protein in the cell medium.

Point 3: The sentence “… Cp is incubated in the CFS from…” should be corrected to … Cp is incubated in the CSF from….

Author’s response: The typo has been amended as requested (page 5 – line 201).

Point 4: Is there any evidence of the expression of αVβ6 in the choroid plexus epithelial cells? Is there any idea of possible integrins expressed in the choroid plexus that could bind to deamidated Cp?

Author’s response: To our knowledge, the αVβ6 integrin heterodimer is not expressed by choroid plexus epithelial cells. The profiling of human primary choroid plexus epithelial cell surface expression of integrin suitable for deamidated Cp binding is reported in the supplementary information of our work quoted as reference number 12 in the manuscript. These data show the absence of αVβ6, αVβ5 but the expression of αVβ3 and α5β1 integrins. To make more clear the phrase that claims that primary choroid plexus epithelial cells do express integrins suitable for Cp binding, we added the following sentence (underlined text) and the reference number 9 (reporting the capability of Cp-ox/de to bind αVβ3 and α5β1): “...Cp-ox/de promotes cell adhesion of primary human choroid plexus epithelial cells (HCPEpiCs), which express aVb3 and a5b1 integrins suitable for isoDGR binding, transducing an intracellular signaling that, as for HaCat, inhibited cell proliferation [9,12], (page 7, lines 283-284).

Point 5: The authors mentioned that the possibility to target choroid plexus epithelial cells might have consequences on the CSF composition. It should be interesting to specify the consequences. CSF turnover?  CSF protein concentration?

Author’s response: We have expanded the section regarding the consequences that targeting choroid plexus epithelial cells might have on CSF composition. Please, find the section in the revised manuscript uploaded, page 7 – lines 288 to 296.

Point 6: The alterations induced in the secretome profile of the choroid plexus epithelial cells after treatment with Cp-ox/de should be discussed in more detail. Treatment with Cp-ox/de increase or decrease the secretion of proteins? Which ones? The treatment induced the release of additional proteins?

Author’s response: We have expanded the paragraph regarding the changing in CPEpiCs secretome profile; please, find the section in the revised manuscript uploaded, page 7 – lines 298 to 307. However, since we have now a submitted manuscript concerning these data, details on the specific differentially expressed proteins have not been included to avoid data disclosure. If deemed mandatory, the manuscript can be sent to this Reviewer in a strictly confidential manner.

Point 7: - The authors should include in the reference list the respective number of each reference mentioned in the manuscript

Author’s response: We apologize for the inconvenience, we have provided for the correction of the mistake.
